# Evolution of Food Trade Networks from a Comparative Perspective: An Examination of China, the United States, Russia, the European Union, and African Countries

**DOI:** 10.3390/foods13182897

**Published:** 2024-09-12

**Authors:** Wei Hu, Dongling Xie, Yilin Le, Ningning Fu, Jianzhen Zhang, Shanggang Yin, Yun Deng

**Affiliations:** 1College of Geography and Environmental Sciences, Zhejiang Normal University, Jinhua 321004, China; huweilt1990@163.com (W.H.); xdl990825@163.com (D.X.); zjz@zjnu.cn (J.Z.);; 2Key Laboratory of Watershed Earth Surface Processes and Ecological Security, Zhejiang Normal University, Jinhua 321004, China; 3Faculty of Geographical Science, Beijing Normal University, Beijing 100875, China; funn1008@163.com

**Keywords:** food trade, comparative analysis, grain trade network, driving factors, QAP analysis

## Abstract

In the intricate landscape of the global food system, a nuanced understanding of dynamic evolution patterns and driving mechanisms of food trade network is essential for advancing insights into the African food trade and maintaining the food security of Africa. This paper constructs a framework for analyzing the food trade network from a comparative perspective by comparing and analyzing the evolution of food trade networks in China, the United States, Russia, the European Union, and African countries. The development trend of food trade between China, Russia, the United States, the European Union, and African countries is relatively good. China, the United States, Russia, and the European Union export far more food to African countries than they import, and bilateral food trade plays an important role in alleviating food supply shortages in Africa. The food trade networks between China, the United States, Russia, the European Union, and African countries exhibit a butterfly-shaped structure centered in Africa, and the overall intensity of bilateral trade linkages is gradually increasing. France has the greatest control over the food trade network between China, the United States, Russia, the European Union, and African countries, and the influence of the United States on the food trade network between China, the United States, Russia, the European Union, and African countries is increasing. China’s independence in the food trade network between China, the United States, Russia, the European Union, and African countries is enhanced, but its control ability is limited. The impact of differences in total population, differences in food production, and geographical borders on the trade network between China, the United States, the European Union, and African countries tends to decrease, while the influence of differences in the proportion of agricultural employment, differences in the arable land available for food production, and institutional distance tends to increase.

## 1. Introduction

In recent years, the international food market has experienced heightened instability due to various factors, such as the COVID-19 pandemic, global climate change, and the Russia–Ukraine conflict. The United Nations Food Programme has issued multiple warnings about the potential impact on the global food supply chain, indicating an overall decreasing trend in the world’s food supply. The decline in the international food supply and the restrictive policies on food export in some countries have exacerbated the volatility of the global food market, resulting in the rise of global food prices. The food imports of a large number of African countries are facing a serious crisis due to the global rise in food prices and fluctuations in the international food market. In this context, food security in African countries has become an urgent issue that the international community urgently needs to pay attention to. On the one hand, in 2020, 98 million people on the African continent suffered from severe food insecurity, accounting for two-thirds of the total global population facing severe food insecurity [1]. On the other hand, armed conflicts, extreme weather, and insufficient food production have exacerbated the contradiction between food supply and demand in Africa, making African countries more dependent on international trade for their food supply. More importantly, the trend of the global concentration of food-producing countries has made food exports from China, the United States, Russia, and the European Union more important in maintaining food security in Africa [2].

China, the United States, Russia, and the European Union have significant impacts on food security in Africa. China is Africa’s largest trading partner and an important importer of African agricultural products. The United States, Russia, and the European Union are the world’s major food producers and important food importers to Africa. As crucial players in food import and export, these nations have intricately woven a complex and ever-evolving food trade network with African countries. While this network has facilitated the global flow of grains and, to some extent, addressed food security challenges, its closeness has simultaneously heightened the vulnerability of grain supply stability. Consequently, the following questions arise: how can African countries ensure food security in their diverse food trade relationships? What factors drive the food trade between China, the United States, Russia, the European Union, and African countries? These questions remain unanswered. This article aims to analyze the evolution and underlying mechanisms of the food trade network between China, the United States, Russia, the European Union, and African countries. By doing so, it seeks to provide valuable insights for enhancing global food security cooperation, particularly in the context of safeguarding food security in Africa.

## 2. Literature Review

In the realm of academia, extensive research has been conducted to explore the characteristics and patterns of food trade networks. It is evident that the strengthening of international trade has led to the globalization of food commodities, with global food production and the number of links in the trade network increasing by more than 50% [3]. The concentration of the world’s grain export market is pronounced, with the United States, Canada, Brazil, Argentina, and Russia serving as the primary exporting countries [4]. Developed countries hold a significant position in the global agricultural trade network, with emerging economies like China, Brazil, and India emerging as vital sources of supply and demand [5]. At the micro level, international agricultural trade manifests a closed, unbalanced, diversified, and multipolar development trend [6]. Research on major staples like corn, rice, and wheat indicates increasing trade connectivity, marked by a significant rise in transaction nodes and linkages. However, the overall global food network remains dominated by core exporting countries [7,8]. The global wheat trade forms a competitive pattern with diminishing competitive relationships over time [9]. The global soybean trade flow has also continued to grow in the past decade, with a multipolar network structure that is increasingly dominated by major soybean-producing countries [10,11]. Corn-trading countries have witnessed improved relationship efficiency, with the United States leading in corn exports, and Japan and South Korea are the primary importers [12]. 

The global food trade network has experienced remarkable changes, and emerging economies, such as China, India, Brazil, and Russia, have become more important, offsetting the historical dominance of other nations, such as the United States, Canada, and Australia [13]. The United States, the United Kingdom, Canada, and South Africa are the core hub countries in the global food trade network, while Central Asian countries and Southeast Asian countries are gradually rising in their position in the global food trade network [14]. The evolution of the global food trade network has shifted from a “single-cluster” to a “multi-cluster” configuration [15]. It is important to note that a country’s position in the food trade network has a broad impact on food security. Countries with high levels of domestic grain production have obvious advantages in terms of natural resources [16]. In terms of competitiveness in agricultural trade and the food trade, countries at the core of the trade network wield stronger control over resources, emphasizing the benefits of open trade for food security and the crucial role of international policy coordination in resisting food crises [17]. The dependency on a small number of exporters and a low diversification of imports lead to a global cereal system vulnerable to shocks [18]. In the context of globalization, a global food trade system dominated by several large countries has increased the food vulnerability of African countries [2]. 

Economic, political, socio-cultural, geographical, and trade agreement factors emerge as crucial drivers affecting the food trade network [19,20,21]. Differences in economic and social development drive closer and more diversified trade networks, while consistency in trade policies promotes sustainability and stability [22]. The interaction of FDI and political stability has a positive and significant impact on food security in Sub-Saharan African countries [23]. Limited arable land, scarce irrigation resources, nutrient-depleted soils, and the impact of variable climates are all contributing to a decline in domestic food production [24,25]. The deterioration of political relations will also reduce the stability of food exports [26]. The Russia–Ukraine conflict has disrupted the structural integrity of international crop trade networks, with the wheat and aggregate food link between Russia and NATO economies declining significantly [27]. However, the global food system’s evolution faces vulnerability due to external pressures, including extreme weather, food reserve levels, speculative effects, rapid economic growth in developing economies, and the depreciation of the dollar, which are all capable of causing systemic damage [28]. 

At present, the global grain market confronts the challenge of diminishing grain supply and declining stocks, exacerbated by various factors, such as labor shortages, frequent grain export restrictions, and escalating world grain prices, which pose great challenges to maintaining food security in Africa [11]. Due to the sensitivity and fragility of natural ecosystems, Africa has caused a series of food security issues [29]. African and Middle Eastern countries primarily rely on trade to meet their food needs, constrained by long-term food shortages exacerbated by climate and water supply challenges [30]. The instability in the international food trade market and Africa’s vulnerable ecological environment contribute to an unfavorable situation for food security on the continent. By 2030, it is expected that the Russia–Ukraine conflict will have a significant impact on the food security of Sub-Saharan Africa, with a contribution of 0.29% [31]. In fact, faced with potential external disruptions, such as armed conflicts, climate change, price fluctuations in global markets, and demographic changes, it is particularly necessary for North African countries to diversify their food imports and increase their resilience to disruptions in supply chains [32].

A complex and interdependent food trade network has developed between Africa, China, the United States, Russia, and the European Union, displaying distinct characteristics in food trade with each country [33]. China has become Africa’s second largest export destination for agricultural products [34]. Bilateral trade between the United States and Africa is declining, but the United States is exerting influence on African countries through aid to Africa [35]. The Russia–Ukraine conflict has complicated trade between Russia and the European Union, but it has also made Africa’s position in Russia’s foreign trade network more important [36]. Although most European Union countries are self-sufficient in agriculture, they still import specific low-substitutability products from Africa [37].

Many studies have analyzed the evolution of food trade networks and the factors affecting the evolution of food trade networks at the global level [5], but the results of the evolution of food networks at the global level are not applicable to the national level. In fact, the food security of African countries is closely related to the food trade, which is especially influenced by food trade with China, the United States, Russia, and the European Union. However, few studies have paid attention to the evolution of food trade networks in African countries, and the differences in food trade linkages between China, America, Russia, the European Union, and African countries have not been clarified. How can we construct a framework for analyzing the food trade network from the perspective of country comparisons? How do we compare and analyze the influence of China, the United States, Russia, and the European Union on the food trade in African countries? How do we compare and analyze the driving factors behind the evolution of the food trade between China, the United States, Russia, the European Union, and African countries? These are urgent issues that need to be addressed to understand the food trade and food security in African countries. Therefore, this paper constructs a framework for analyzing the food trade network from the perspective of country comparisons and analyzes the evolution of the food trade network between China, the United States, Russia, Europe, and African countries and their driving factors.

The structure of this article is as follows. Section 1 provides an introduction to the research background of the food trade network between China, the United States, Russia, the European Union, and African countries. Section 2 is the literature review. Section 3 outlines the research methods and the data sources employed in this study. Section 4 delves into the analysis of the evolution of the food trade network between China, the United States, Russia, the European Union, and African countries. Section 5 discusses the driving factors of the evolution of the food trade network between China, the United States, Russia, the European Union, and African countries. Section 6 and Section 7 present the discussion and the conclusions, respectively.

## 3. Materials and Methods

### 3.1. Social Network Analysis Method

Social network analysis is a research methodology that concentrates on elucidating the positions and interrelationships among actors within a network, thus offering insights into its structure [10]. Social network analysis can not only reveal the evolution characteristics of food networks between China, the United States, Russia, the European Union, and African countries but also compare and analyze the differences in food network connections between China, the United States, Russia, the European Union, and African countries. This article employs the social network analysis method to illuminate the comprehensive and structural characteristics of the food trade network involving China, the United States, Russia, the European Union, and African countries. Key indicators, such as network density, the average degree, and the clustering coefficient, are utilized for this purpose. Additionally, the impact of core countries on the food trade network is examined and compared through indicators like the weighted degree, intermediary centrality, and proximity centrality. The indicators of the social network analysis are shown in Table 1.

### 3.2. QAP Analysis

QAP analysis (Quadratic Assignment Procedure) is a method used to analyze social network data. It performs regression analysis by comparing the similarity of the grid values of two matrices, and it provides the correlation coefficient between the two matrices [5]. QAP analysis does not require variables to be independent of each other, and it does not need to consider the issue of multicollinearity among influencing factors, making the regression analysis of relational data more stable [38]. Therefore, this study selected QAP analysis to analyze the impact of driving factors on the food trade network between China, the United States, Russia, the European Union, and African countries.

### 3.3. Data Source

The scope of food is very broad, but cereals are at the core of the grain trade. Referring to relevant research [14], this article constructs a food trade network between China, the United States, Russia, the European Union, and African countries through the cereals trade. The cereals trade data are derived from the United Nations Comtrade database (https://comtradeplus.un.org/), and the food price index is derived from the Food and Agriculture Organization of the United Nations (https://www.fao.org/). Geographical borders and the geographical distance between capitals are derived from the CEPII (http://www.cepii.fr). Food production, the arable land available for food production, the proportion of the agricultural employment, the per capita GDP, renewable freshwater resources per capita, and the worldwide governance indicators are obtained from the World Bank (https://data.worldbank.org/). Additionally, the total population data are sourced from the Food and Agriculture Organization of the United Nations (https://www.fao.org/).

The study area encompasses China, the United States, Russia, the European Union, and 54 African countries. Considering the UK’s formal Brexit on 31 January 2020, the study incorporates the UK within the European Union alongside the 27 existing European Union member states.

## 4. Evolution of Food Trade Networks between China, the United States, Russia, the European Union, and African Countries

### 4.1. Evolution of the Food Trade between China, the United States, Russia, the European Union, and African Countries

From 2001 to 2021, the total food trade volume between China, the United States, Russia, the European Union, and African countries increased from USD 2.662 billion to USD 11.067 billion, with a relatively fast overall growth (Figure 1). Affected by various factors, such as climate change, food production, and supply and demand, the food price index fluctuated from 51.8 to 131.2 from 2001 to 2021. From the perspective of stage changes, the total food trade volume between China, the United States, Russia, the European Union, and African countries is basically consistent with the trend of food price index changes, both of which showed significant turning points in 2008, 2011, and 2021. According to the evolution trend of the food trade volume and the food price index, the evolution of food in China, the United States, Russia, the European Union, and African countries can be divided into four stages. Slow growth phase (2001–2005): During this phase, the total food trade volume increased gradually from USD 2.662 billion to USD 3.726 billion, and the food price index rose from 51.8 to 60.8, reflecting a period of stable market development. Food crisis phase (2006–2012): This phase was marked by food crises in 2007–2008 and 2010–2011, with the food price index peaking at 137.6 in 2008 and 142.2 in 2011. The food crisis brought about turbulence in the supply and demand of food, resulting in significant fluctuations in the food price index during this phase. Steady decline phase (2013–2019): During this period, both the total food trade volume and the food price index showed reduced volatility. The food price index declined from 129.1 to 96.6, while the total food trade volume fell slightly from USD 8.697 billion to USD 8.560 billion. Although the food market gradually stabilized, a divergence emerged between the total food trade volume and the food price index, suggesting a decoupling of trends. Rapid growth phase (2020–2021): Global climate change, the Russia–Ukraine conflict, and the COVID-19 pandemic have jointly pushed the food price index to rapidly increase from 103.1 to 131.2, while the total food trade volume increased from USD 10.023 billion to USD 11.067 billion.

From 2001 to 2021, the total food trade volume between China and African countries rose from USD 158 million to USD 446 million, largely driven by China’s growing exports to Africa. During the same period, the total food trade volume between the United States and African countries exhibited a modest decline, decreasing from USD 1.416 billion to USD 1.408 billion, primarily driven by fluctuations in the United States food exports to Africa. From 2001 to 2021, the total food trade volume between Russia and African countries surged from USD 440 million to USD 3.529 billion. In the 21st century, as the Russian economy recovered and agricultural production efficiency improved, Russia prioritized food trade with African nations. Actively engaging in trade cooperation, Russia supplied grains and other agricultural products to African countries, resulting in a fluctuating upward trend in Russian food exports to Africa. From 2001 to 2021, the total food trade volume between the European Union and African countries surged from USD 1.043 billion to USD 5.684 billion. Both the EU’s food exports to African countries and food imports from African countries demonstrated a fluctuating upward trend. European Union food exports to Africa increased from USD 1.015 billion to USD 5.596 billion, while European Union food imports to Africa rose from USD 28 million to USD 88 million. It is not difficult to find that China, the United States, Russia, and the European Union have far more food exports to African countries than imports, highlighting the role of bilateral food trade in mitigating food supply shortages in Africa.

### 4.2. Characteristics of Food Trade Networks between China, the United States, Russia, the European Union, and African Countries

The food trade linkages among China, the United States, Russia, the European Union, and African countries have formed a dynamic and complex network structure. To analyze the evolution of this food trade network, this study uses 2001, 2011, and 2021 as reference points for comparison.

The food trade network, forming a butterfly-shaped structure centered around Africa, illustrates increasing interconnection due to intensified trade activities (Figure 2). Over this period, the number of trade linkages in the food trade network has risen from 1375 to 1808, accompanied by an increase in the average degree from 16.369 to 21.271, indicating both an expansion in the number of trade linkages and the complexity of the network. Examining network density, there is an upward trend from 0.197 to 0.253 during 2001–2021, signaling an enhanced closeness in food trade relationships among these countries. Analyzing the average weighted degree reveals substantial growth from 239,414,216.4 to 1,017,032,838, indicating a clear upward trajectory and increased intensity of trade linkages. Regionally, the European Union and African countries exhibit frequent food trade, China and African countries witness an augmentation in food trade linkages and strengthened connections, and Russia maintains a positive food trade relationship with African countries, while the strength of food trade linkages between the United States and African countries shows an overall decline. Over the period of 2001–2021, the average clustering coefficient of the food trade networks has risen from 0.562 to 0.626, signifying an enhancement in network cohesion. Simultaneously, the average path length has decreased from 1.938 to 1.804, suggesting improved accessibility and closer food trade ties between countries. The increasing average clustering coefficient and decreasing average path length further indicate an enhancement in the small-world characteristics of the network.

### 4.3. The Characteristics of Food Trade Networks’ Structure between China, America, Russia, the European Union, and African Countries

Node size is positively correlated with the core degree, while the node’s color signifies the country’s position in the Core–Periphery structure. Red indicates a core country, green denotes a semi-peripheral country, and orange represents a peripheral country. Core countries have a core degree exceeding 0.1, semi-peripheral countries fall within the 0.01–0.1 range, and peripheral countries have a core degree below 0.01. The food trade networks between China, the United States, Russia, the European Union, and African countries exhibit a distinct “Core-Periphery” structure. Figure 3 illustrates a decrease in the number of core and peripheral countries, accompanied by a rise in the number of semi-peripheral countries, reaching 56 countries by 2021.Throughout the study years, the core degree rankings of most countries experience considerable fluctuations, indicating relative instability. France, Germany, and the Netherlands consistently maintain a core degree above 0.1, securing a core position in the food trade networks. China rapidly ascends to the top core degree ranking, while Russia progresses from a peripheral to a semi-peripheral country. By 2021, the United States’s core degree rises to the third position, indicating an enhanced position in the food trade networks. Conversely, Spain and Italy witness a noticeable decline in their core degrees, with both leaving the core circle in 2021, signifying diminishing importance. Slovenia, Cyprus, and other countries consistently hold peripheral positions in the food trade network.

In the food trade networks encompassing China, the United States, Russia, the European Union, and African countries, the majority of African countries fall into the semi-peripheral or peripheral categories, characterized by low core degrees. Algeria’s core degree exhibits an initial increase followed by a decrease, entering the core country range in 2011. Thirteen countries, including Egypt, Central Africa, Equatorial Guinea, and Nigeria, consistently belong to semi-peripheral countries, while South Africa, Mauritius, and Kenya remain peripheral countries, occupying an outer position in the food trade network. Notably, Egypt, Algeria, Equatorial Guinea, and Nigeria achieve higher rankings in terms of core degree, securing elevated positions in the food trade networks among African countries.

Cohesive subgroup analysis reveals the relationships between nodes in the food trade networks, illustrating that closely related nodes form subgroups (Figure 4). Subgroup I comprises main countries, such as China, the United States, and European Union countries like France, Belgium, the Netherlands, Germany, Spain, the United Kingdom, Italy, and Russia. The density within subgroup I increased from 0.628 to 1 during 2001–2021, ranking first among the subgroups and indicating the closest food trade relationships. Subgroup II, the largest subgroup in terms of countries, is dominated by EU nations, such as Luxembourg, Ireland, Estonia, Slovenia, and Austria, and North African countries, including Algeria, Libya, Tunisia, and Morocco. This subgroup consistently includes European Union countries like Luxembourg, Ireland, and Estonia, demonstrating relatively close food trade relationships with North African countries. Countries in subgroup III are mainly East African countries like Malawi, Rwanda, Burundi, Kenya, and Zambia, and South African countries, such as Namibia, Swaziland, Lesotho, and Botswana. Subgroup IV is characterized by West African countries, like Senegal, Togo, Burkina Faso, Côte d’Ivoire, Mali, Benin, and Niger, and Central African countries, including Chad, Gabon, Cameroon, and the Republic of Congo. The density within subgroup IV decreased from 0.301 to 0.13 during 2001–2021, indicating a decline in food trade linkages among subgroup IV member countries.

In summary, subgroup III and IV consist primarily of African countries, indicating stronger cohesion within African countries in terms of food trade. However, subgroup I and II, with fewer African countries, still play crucial roles in the food trade networks. Subgroup I, in particular, maintains close relationships with all other subgroups. From 2001 to 2021, the connection between subgroup I and II became the closest, with the density increasing from 0.749 to 0.935. The density between subgroup I and IV increased from 0.475 to 0.668, and that between subgroup I and III reached 0.622. This underscores subgroup I as the core subgroup, with main countries, such as China, the United States, and European Union countries, enhancing the closeness of food trade relationships with African countries. Additionally, the number of African countries included in subgroup II in-creased from 2001 to 2021, indicating improved food trade relationships between Luxembourg, Ireland, Estonia, Slovenia, Austria, and African countries and increased cohesion with African countries.

### 4.4. Influence of Core Countries on Food Trade Networks

Indicators, such as the weighted degree, the weighted out degree, the weighted in degree, closeness centrality, and intermediary centrality, provide insights into the impact of various countries on the food trade network between China, the United States, Russia, the European Union, and African countries from different perspectives. The weighted degree refers to the sum of weights of the edges connected to a node, reflecting the scale of food trade of each country. The weighted out degree and weighted in degree can reflect the food export status and food import status of each country, respectively. Closeness centrality can be used to measure the extent to which a country is not controlled by other countries, reflecting the independent capacity of countries in the food trade network. Intermediary centrality measures the extent to which a country is able to act as a “middleman”, providing linkages between other countries and reflecting the ability of countries to control the food trade. By comparing the rankings of the weighted degree, closeness centrality, intermediary centrality, and proximity centrality in 2001 and 2021, it can be found that France, the United States, Germany, Belgium, and the Netherlands have always ranked among the top 10 in the food trade network, indicating that these five countries have a greater influence on the food trade network.

From 2001 to 2021, France had the highest weighted degree and intermediary centrality among all countries, and its closeness centrality ranked among the top in the food trade network. This indicates that France has the greatest control over the food trade network between China, the United States, Russia, European Union countries, and African countries, occupying a pivotal position in the food trade network. The United States showed a continuous increase in its weighted degree and maintained a top-three position in intermediary centrality, with its closeness centrality rising from 0.912 to 0.955. This indicates that the United States can directly form food trade linkages with most African countries, and its influence on the food trade network continues to increase. Germany experienced declines in its weighted degree, closeness centrality, and intermediary centrality, signaling a reduction in its influence within the food trade network. The scale of food trade between the Netherlands, Belgium, and African countries is relatively small, but the closeness centrality and intermediary centrality between the two countries and African countries have shown a slight upward trend, indicating an increase in their independence and intermediary nature in the food trade network. China’s weighted degree in the food trade network rose to the first place in 2021, which shows that China has gradually become the core country in the food trade network of China, the United States, Russia, the European Union, and African countries. China’s closeness centrality in the food trade network is constantly improving, but its intermediary centrality is always at a low level. This indicates that China’s independence in the food trade network between China, the United States, Russia, the European Union, and African countries is enhanced, but its control ability is limited. Russia’s weighted degree, closeness centrality, and intermediary centrality in the food trade network continue to increase, but its closeness centrality and intermediary centrality are still at a low level, which indicates that Russia has significant room to improve its influence on the food trade network between China, the United States, Russia, the European Union, and African countries.

## 5. Driving Factors of the Evolution of the Food Trade Networks between China, the United States, Russia, the European Union, and African Countries

### 5.1. Model Construction

Cultivated land resources, scientific and technological levels, the economic level, institutional factors, social and cultural factors, psychological factors, trade agreements, and political relations are all factors that affect the evolution of the food trade networks between China, the United States, Russia, the European Union, and African countries [23]. However, psychological factors, scientific and technological levels, and political relations do not directly affect the evolution of the food trade networks between China, the United States, Russia, the European Union, and African countries. They indirectly affect the evolution of the food trade networks between China, the United States, Russia, the European Union, and African countries. More importantly, data on psychological factors, scientific and technological levels, political relations, and other related indicators of African countries cannot be obtained at present. Drawing upon relevant studies on the driving factors of food trade networks [39] and others, and considering data availability, nine factors are identified as key driving factors in the evolution of food trade networks between China, the United States, Russia, the European Union, and African countries. These factors include differences in per capita GDP, total population, the proportion of agricultural employment, food production, the availability of arable land for food production, renewable freshwater resources per capita, geographical distance between capitals, geographical borders, and institutional distance (Table 2).

Taking into account data availability, the article selects the years 2001, 2011, and 2021 as the time nodes, defines the research scope to include China, the United States, Russia, EU countries, and African countries (excluding South Sudan), and employs the matrix of food trade volume between these entities (Wi) as explanatory variables. The explanatory variables are presented in a matrix networks data format categorized into two types, difference matrix and binary matrix, with the exception of the matrix representing geographical distance between capitals. The geographical borders matrix is a binary matrix, assigning a value of 1 if a land border exists between two countries and 0 otherwise. Differences in per capita GDP, total population, the proportion of agricultural employment, food production, arable land area, renewable freshwater resources per capita, and institutional distance all belong to the difference matrix. Except for institutional distance, the difference matrix of other variables is obtained by subtracting the corresponding indicator data of each country and taking the absolute value. Institutional distance is the difference between countries in rules, norms, regulations, and cognition [40], and its difference matrix can be obtained by calculating the worldwide governance indicators [41]. The difference matrices, encompassing differences in per capita GDP, total population, the proportion of agricultural employment, food production, arable land area, renewable freshwater resources per capita, and institutional distances, are derived by subtracting corresponding data for each country and taking absolute values. Additionally, to enhance the model’s fit and to mitigate the influence of different dimensions, the matrices representing the food trade volume of explained variables, the difference in explanatory variables, and the geographical distance between capitals are logarithmically transformed. Based on this analysis, the QAP model is formulated as follows:$$\ln\left(W_{i}+1\right)=\beta_{0}+\beta_{1}CONTIG+\beta_{2}\ln\left(CEREAL+1\right)+\beta_{3}\ln\left(LAND+1\right)+\beta_{4}\ln\left(REN+1\right)+\beta_{5}\ln\left(INSD+1\right)\;+\beta_{6}\ln\left(P.GDP+1\right)+\beta_{7}\ln\left(WATER+1\right)+\beta_{8}\ln\left(POPU+1\right)+\beta_{9}\ln\left(DISTCAP+1\right)+\mu_{i}$$

### 5.2. Results of QAP Analysis

We employ UCINET to analyze the food trade networks of China, the United States, Russia, the European Union, and Africa, considering all explanatory variables for the years 2001, 2011, and 2021. With the exception of the difference in renewable freshwater resources per capita, the remaining eight explanatory variables have demonstrated significance in these three years. Notably, differences in the proportion of agricultural employment, geographical distance between capitals, and institutional distance exhibit a negative correlation with food trade networks, while geographical borders, differences in food production, differences in the arable land available for food production, differences in per capita GDP, and differences in total population show a positive correlation. The difference in renewable freshwater resources per capita has achieved statistical significance at the 10% level in the food trade networks of 2001 and 2011. However, the correlation analysis results for 2021 are not statistically significant, and the correlation coefficients for all three years are low. This suggests a weak correlation between the difference in renewable freshwater resources per capita and the food trade networks.

We utilize UCINET to conduct a comprehensive regression analysis of explanatory variables and the food trade networks of China, the United States, Russia, the European Union, and African countries for the years 2001, 2011, and 2021 (Table 3). The overall model-fitting degree with adjusted coefficients of determination hovers around 40%. The regression models for these three years generally pass the significance test, suggesting that they effectively explain the linkages within food trade networks. Except for the difference in renewable freshwater resources per capita, all explanatory variables achieved significance across the three years. Notably, in both 2001 and 2021, seven variables are significant at the 1% level and one is significant at the 5% level, while, in 2011, eight variables are significant at the 1% level and one is significant at the 10% level. This analysis affirms the accuracy of the regression results and underscores the reasonableness of variable selection in the model. In summary, the constructed regression model in this article exhibits robust explanatory power, with significant results for the explanatory variables. The selection of these variables is deemed reasonable.

The positive drivers influencing the evolution of food trade networks include the difference in per capita GDP, differences in total population, differences in the arable land available for food production, geographical borders, and differences in food production. Notably, the difference in per capita GDP plays a pivotal role. Greater economic development differences facilitate the establishment and expansion of food trade linkages. The regression coefficient in 2021 indicates that a 1% increase in the difference in per capita GDP leads to a 0.393% increase in food trade volume. Differences in total population also impact food trade positively, with a 0.198% increase in food trade volume for every 1% rise in differences in the total population. Differences in the arable land available for food production contribute to the complementarity of resources, promoting food trade. Geographical borders, especially with common land borders, facilitate trade by reducing transportation costs. However, the influence of differences in food production on food trade networks diminishes gradually.

Conversely, geographical distance between capitals, differences in the proportion of agricultural employment, institutional distance, and the difference in renewable freshwater resources per capita exert significant negative impacts on food trade networks. Greater geographical distance increases transportation costs, making food trade relationships less likely. The influence of geographical distance between capitals has slightly increased from 2001 to 2021, with a 0.286% decrease in food trade volume for every 1% increase in influence in 2021. The proportion of agricultural employment reflects the position of agriculture in the national economy and the national industrial structure. The greater the difference in the proportion of agricultural employment between countries, the greater the difference in food production and import and export, and the lower the probability of food trade between countries [42]. Therefore, larger differences in the proportion of agricultural employment hinder the food trade. Institutional distance’s negative impact on food trade relationships has intensified over time. In contrast, the difference in renewable freshwater resources per capita, while negative, has the smallest influence on food trade networks. Both the regression coefficients and the correlation coefficients for the difference matrix are not statistically significant, indicating that this factor is not a primary driving force in food trade networks.

## 6. Discussion

The food trade network analysis framework constructed in this paper from the perspective of country comparisons effectively reveals the differences in the food trade linkages between China, the United States, Russia, the European Union, and African countries, which indicates that the grain trade network analysis from the perspective of country comparisons can reveal the evolution laws of the food trade at a more micro level and more accurately. In the era of globalization, the position of a country in the food trade network to some extent affects its food security, and clarifying the interdependence of countries in the food trade network is particularly important. However, using only overall network characteristics or individual network attributes is not sufficient to fully grasp the interdependence of countries in the food trade. The food trade network analysis from a comparative perspective can not only reveal the interdependence and influence of food trade among countries but also more clearly identify the differences in attribute characteristics among countries in the food trade network. The comparative analysis framework of food trade networks constructed in this paper will provide a reference for comparing the attribute characteristics and interdependence of countries in the trade network in the future. In addition, in light of food security becoming a core national security concern, the article emphasizes the importance of considering both structural characteristics and key driving factors in food trade networks, which will help us better understand the causal mechanism of the evolution of the food trade network.

From the perspective of the country’s position in the food trade network, France has always held a core position in the food trade network between China, the United States, Russia, the European Union, and African countries, while China’s position in the food trade network between China, the United States, Russia, the European Union, and African countries has significantly increased. African countries, such as Egypt, Algeria, and Nigeria, have a high centrality in the food trade network, but their position in the food trade network is due to high food import flows, and their dependence on food-exporting countries, such as France, the United States, and Russia, is higher. From the perspective of the structure of the food trade network, the weighted centrality degree of France, China, and the United States in the food trade network between China, the United States, Russia, the European Union, and African countries has decreased during 2001–2021. This indicates that the “Core-Periphery” structural characteristics of the food trade network between China, the United States, Russia, the European Union, and African countries have weakened. From the perspective of a cohesive subgroup in the food trade network, the regional characteristics of African countries are relatively obvious, and the degree of fragmentation of the subgroup of African countries is much greater than that of non-African countries, such as China, the United States, France, Germany, and Russia. From the perspective of the influence of core countries in the food trade network, China, the United States, and Russia have significantly increased their influence in the food trade network between China, the United States, Russia, the European Union, and African countries. In the future, these countries will have a greater influence on the food security of African countries.

The driving mechanism of the evolution of the food trade network between China, the United States, Russia, the European Union, and African countries indicates that economic factors, population factors, natural resource endowments, geographical distance, and institutional differences all have an impact on the food trade, which in turn affects the food supply of African countries. More importantly, when these factors are combined with potential external disturbances, such as climate change, global market price fluctuations, and geopolitical conflicts, they will have a significant impact on the food security of African countries [32]. Therefore, it is particularly necessary to adopt various measures to maintain the stability of the food supply in African countries. On the one hand, African countries should increase the diversity of their food-trading partners and avoid relying solely on a few countries for food imports. On the other hand, African countries should increase their own food production and enhance the resilience of their food supply by expanding their food cultivation area and improving their food production technology. Given that China, the United States, Russia, and France all have significant impacts on the food trade in African countries, African countries can strengthen cooperation with them in various areas, such as food production, food supply chains, and agricultural production technologies, to enhance food security.

In contrast to prior studies, this article adopts a micro perspective focusing on China, Russia, the European Union, and African countries. It identifies butterfly-shaped morphological characteristics and a Core–Periphery structure in the food trade network, thus presenting a nuanced departure from the evolution patterns observed in Belt and Road countries [16]. This approach provides clearer insights into the cohesive effects of core subgroups on African countries. Regarding driving factors, this article compares and analyzes the dynamic changes in the impact of driving factors, such as differences in per capita GDP, geographical distance, and food production, on the food trade volume between China, the United States, the European Union, and African countries, enabling us to better understand the driving effects of different driving factors on the food trade network.

Political relations are an important factor affecting food trade between countries [26]. Due to the unavailability of data on political relations between China, the United States, Russia, the European Union, and African countries, this article did not analyze the impact of political relations on the food trade network between China, the United States, Russia, the European Union, and African countries. Although this paper emphasizes the impact of global events, such as the Russia–Ukraine conflict and the COVID-19 pandemic, on the food trade network between China, the United States, Russia, the European Union, and African countries, this paper does not accurately identify the extent of the impact of these geopolitical events on the food trade network between China, the United States, Russia, the European Union, and African countries. In fact, the development of the food trade network is a dynamic and evolving process, and it is particularly necessary to predict the evolutionary trend of the food trade network between China, the United States, Russia, the European Union, and African countries. In the future, we should strengthen research on the impact of political relations and global events on the food trade network between China, the United States, and the European Union and conduct multi-scenario simulations of the evolution of the food trade network between China, the United States, Russia, the European Union, and African countries.

## 7. Conclusions

We constructed a framework for analyzing the food trade network from the perspective of country comparisons and analyzed the evolution of the food trade network between China, the United States, Russia, the European Union, and African countries and their driving factors by using social network analysis and QAP analysis. The development trend of the food trade between China, Russia, the United States, the European Union, and African countries is relatively good. The United States is a relatively important food-trading partner for Africa, but the food trade volume between the United States and African countries has tended to decrease. China, the United States, Russia, and the European Union export far more food to African countries than they import, and bilateral food trade plays an important role in alleviating food supply shortages in Africa. The number of trade linkages, the network density, and the weighted degree of the food trade network between China, the United States, Russia, Europe, and African countries are all increasing, and the overall intensity of bilateral trade linkages is gradually increasing. The food trade networks between China, the United States, Russia, the European Union, and African countries exhibit a distinct “Core-Periphery” structure. The food trade network between China, the United States, Russia, the European Union, and African countries can be divided into four subgroups, but the cohesion of the food trade within African countries is stronger. France has the greatest control over the food trade network between China, the United States, Russia, the European Union, and African countries, and the influence of the United States on the food trade network between China, the United States, Russia, the European Union, and African countries is increasing. China’s independence in the food trade network between China, the United States, Russia, the European Union, and African countries is enhanced, but its control ability is limited. The impact of differences in the total population, differences in food production, and geographical borders on the trade network between China, the United States, the European Union, and African countries tends to decrease, while the influence of differences in the proportion of agricultural employment, differences in the arable land available for food production, and institutional distance tends to increase.

## Figures and Tables

**Figure 1 foods-13-02897-f001:**
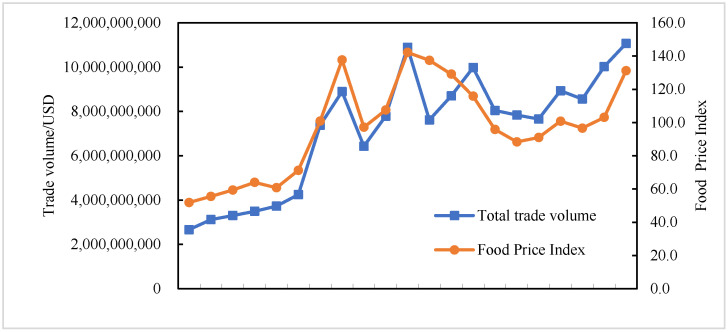
The total food trade volume and changes in the world food price index between China, the United States, Russia, the European Union, and African countries from 2001 to 2021.

**Figure 2 foods-13-02897-f002:**
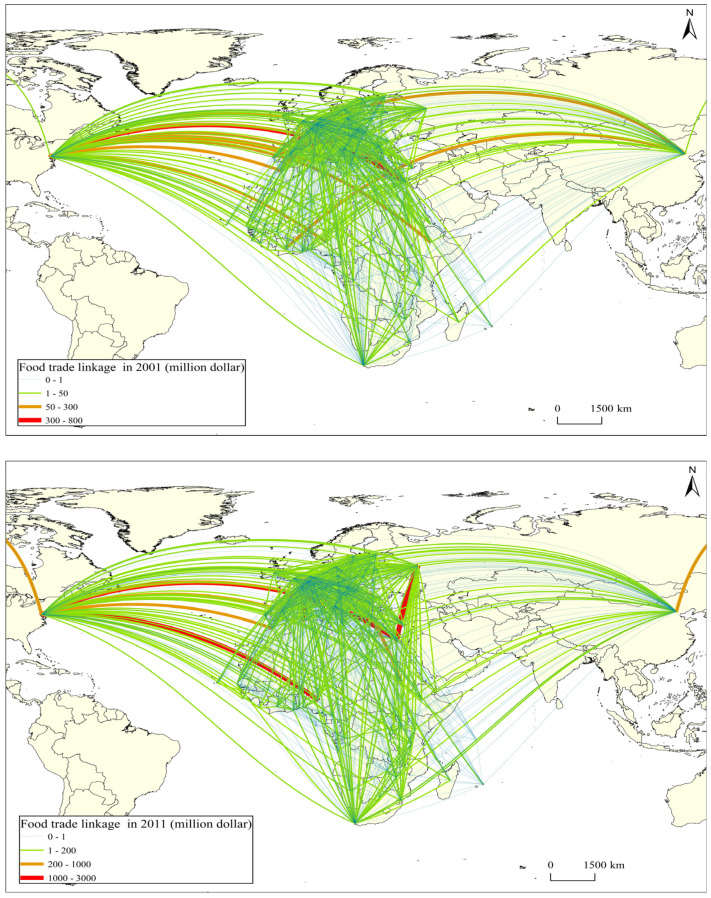
Food trade linkage between China, the United States, Russia, the European Union, and African countries. Note: the map is drawn based on the standard map of GS (2016) 2931 that comes from the standard map service system of the Ministry of Natural Resources of the People’s Republic of China (http://bzdt.ch.mnr.gov.cn/index.html, accessed on 9 September 2024).

**Figure 3 foods-13-02897-f003:**
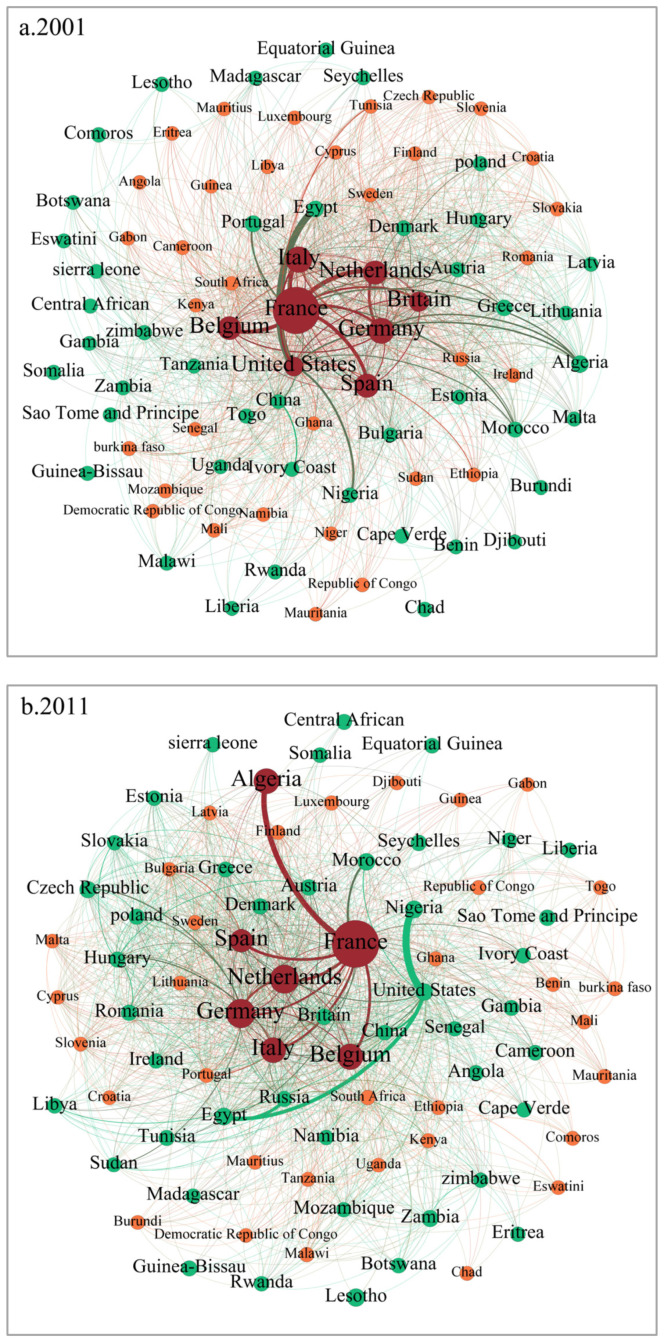
The Core–Periphery structure of food trade networks between China, the United States, Russia, the European Union, and African countries in 2001, 2011, and 2021.

**Figure 4 foods-13-02897-f004:**
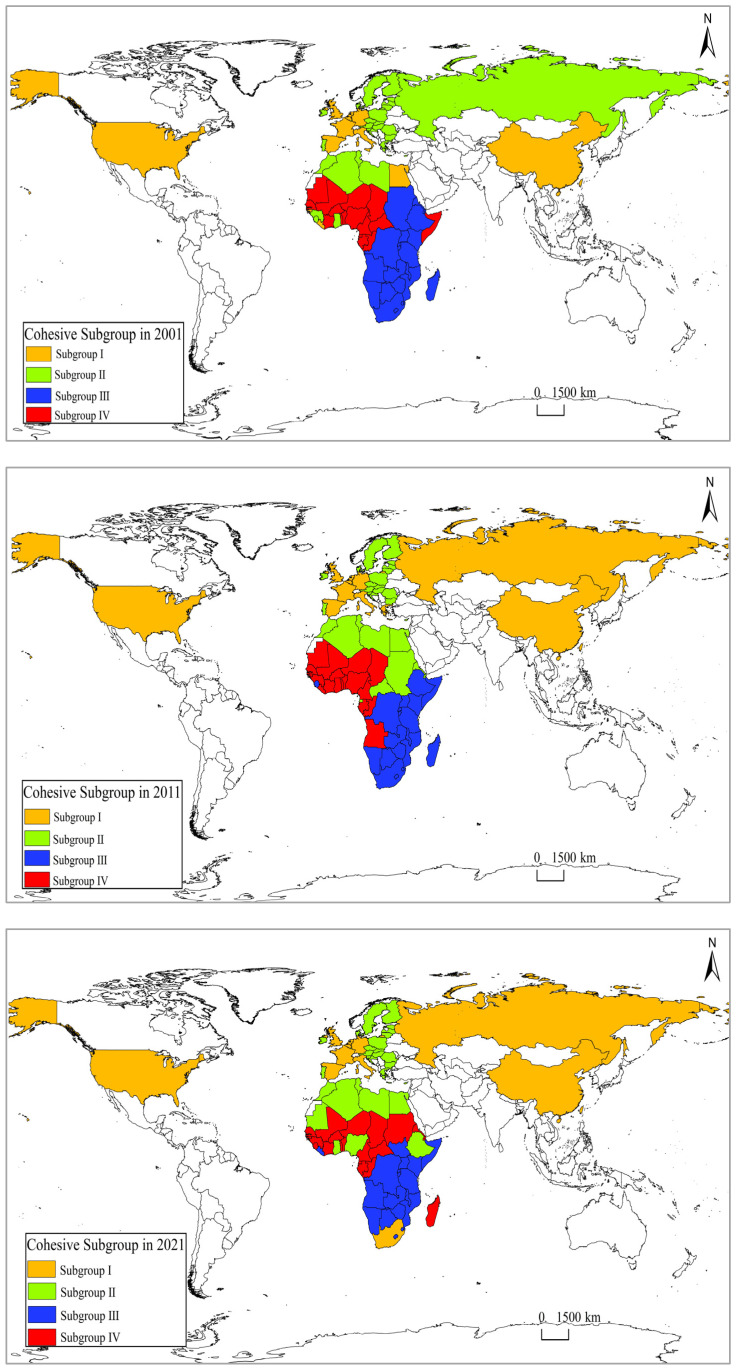
Cohesive subgroup of the food trade networks between China, the United States, Russia, the European Union, and African countries in 2001, 2011, and 2021. Note: The map is drawn based on the standard map of GS (2016) 2931 that comes from the standard map service system of the Ministry of Natural Resources of the People’s Republic of China (http://bzdt.ch.mnr.gov.cn/index.html, accessed on 9 September 2024).

**Table 1 foods-13-02897-t001:** The indicators of the social network analysis.

Indicators	Forms	Connotation
Weighted degree (*D_i_*)	$D_{i}=\sum_{j=1}^{n}w_{ij}$	In the formula, wij represents the trade volume between countries i and j, and n represents the number of countries that have trade relations with country i.
Weighted in degree (*D_i_^in^*)	$D_{i}^{in}=\sum_{j=1}^{n}w_{ij}$	In the formula, Wij represents the import trade volume between country i and country j, and n represents the number of countries that have trade relations with country i.
Weighted out degree (*D_i_^out^*)	$D_{i}^{out}=\sum_{j=1}^{n}w_{ij}$	In the formula, Wij represents the export trade volume between country i and country j, and n represents the number of countries that have trade relations with country i.
Network density (*M*)	$M=\frac{2m}{n(n-1)}$	In the formula, m represents the actual number of relationships in the network, and n represents the number of nodes in the network.
Clustering coefficient (*C*)	$C=\frac{1}{n}\sum_{i=1}^{n}\frac{2M_{i}}{k_{i}(k_{i}-1)}$	In the formula, n represents the number of nodes, ki represents the number of adjacent nodes of node i, and Mi represents the actual number of edges between adjacent nodes of node i.
Closeness centrality (*CC_i_*)	$CC_{i}=\frac{n-1}{\sum_{j=1,\;j\neq i}d_{ij}}$	In the formula, n represents the number of nodes in the network, and d represents the shortcut distance between node i and node j.
Intermediary centrality (*BC_i_*)	$BC_{i}=\frac{2\sum_{j}^{n}\sum_{k}^{n}b_{jk}(i)}{n^{2}-3n+2}$	In the formula, bjk (i) represents the ability of the third node i in the trade network to control the association between j and k, and n represents the number of nodes in the network.

**Table 2 foods-13-02897-t002:** Explanatory variables and descriptions.

Variable	Implication	Symbol
Geographical borders	Binary matrix of geographical borders	CONTIG
Differences in food production	Difference matrix of food production	CEREAL
Differences in the arable land available for food production	Difference matrix of the arable land available for food production	LAND
Differences in the proportion of agricultural employment	Difference matrix of the proportion of agricultural employment	REN
Institutional distance	The matrix of institutional distance	INSD
The difference in per capita GDP	Difference matrix of per capita GDP	P.GDP
The difference in renewable freshwater resources per capita	The difference matrix of renewable freshwater resources per capita	WATER
Differences in total population	Population difference matrix	POPU
Geographical distance between capitals	The matrix of geographical distance between capitals	DISTCAP

**Table 3 foods-13-02897-t003:** Results of QAP regression analysis.

Explanatory Variable	2001	2011	2021
Geographical borders	0.124 ***	0.132 ***	0.093 ***
Differences in food production	0.111 ***	0.096 ***	0.042 **
Differences in the arable land available for food production	0.093 ***	0.112 ***	0.135 ***
Differences in the proportion of agricultural employment	−0.196 ***	−0.238 ***	−0.220 ***
The difference in per capita GDP	0.303 ***	0.312 ***	0.393 ***
The difference in renewable freshwater resources per capita	−0.039	−0.043 *	−0.017
Differences in total population	0.265 ***	0.227 ***	0.198 ***
Geographical distance between capitals	−0.285 ***	−0.313 ***	−0.286 ***
Institutional distance	−0.064 **	−0.097 ***	−0.152 ***
AJ-R^2^	0.371	0.403	0.378
Model significance	0.000	0.000	0.000
observed value	6972	6972	6972

Note: *** means *p* ≤ 0.01, ** means 0.01 < *p* ≤ 0.05, * means 0.05 < *p* ≤ 0.1.

## Data Availability

The original contributions presented in the study are included in the article, further inquiries can be directed to the corresponding author.

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
