# Peer review of "Evolution of Food Trade Networks from a Comparative Perspective: An Examination of China, the United States, Russia, the European Union, and African Countries"

_foods, 2024, doi:10.3390/foods13182897_

Round 1
Reviewer 1 Report
Comments and Suggestions for Authors
The paper titled “Evolution of Grain Trade Network from a Comparative Perspective: An Examination of China, the United States, Russia, the European Union, and African Countries” is engaging. However, I have some suggestions and recommendations to further enhance it.
1. This study provides a fascinating comparison of food trade networks. However, one question arises: why does the study focus exclusively on these specific countries?
2. The study should include a background on the food market and its fluctuating behavior to further strengthen the arguments presented.
3. Psychosocial factors also impact trade networks. Therefore, it is recommended to discuss these factors in the research.
4. The Quadratic Assignment Procedure (QAP) is an extension of the Mantel test within a regression framework, developed in the field of psychometrics.
5. Thus, it is necessary to investigate psychosocial factors as they can aid in the analysis of interrelated items and contribute to social network analysis.
6. The conclusion and discussion should be presented under separate headings.
7. The discussion should be linked to the trade networks of food in the specified countries.
Author Response
Dear reviewer:
Thank you very much for your careful guidance on the paper. Your feedback has greatly helped improve the quality of our paper, and we have greatly benefited from it. According to your comments, we have made a comprehensive revision of the paper. The specific revision parts have been marked in blue in the revised manuscript. We truly appreciate you again.
Comments 1: This study provides a fascinating comparison of food trade networks. However, one question arises: why does the study focus exclusively on these specific countries?
Reponses 1: Thank you very much for your question. First, in 2020, 98 million people on the African continent suffered from severe food insecurity, accounting for two-thirds of the total global population facing severe food insecurity. Second, armed conflicts, extreme weather and insufficient food production have exacerbated the contradiction between food supply and demand in Africa, making African countries more dependent on international trade for food supply. More importantly, the trend of global concentration of food producing countries has made food exports from China, the United States, Russia and the European Union more important in maintaining food security in Africa. Third, China, the United States, Russia, and the European Union have significant impacts on food security in Africa. China is Africa's largest trading partner and an important importer of African agricultural products. The United States, Russia and the European Union are the world's major food producers and important food importers to Africa. For these reasons, this paper chooses the food trade between China, America, Russia, the European Union and African countries as a case study.
Comments 2:The study should include a background on the food market and its fluctuating behavior to further strengthen the arguments presented.
Reponses 2: Your comments are very helpful and thank you very much for your guidance. According to your advices, we have added the background of the food market and its fluctuating behavior in the revised manuscript.
Comments 3: Psychosocial factors also impact trade networks. Therefore, it is recommended to discuss these factors in the research.
Reponses 3: Thanks very much for your advice. We have added an explanation of the selection of driving factors in the revised manuscript. Cultivated land resources, scientific and technological level, economic level, institutional factors, social and cultural factors, psychological factors, trade agreements and political relations are all factors that affect the evolution of the food trade networks between China, the United States, Russia, European Union and African countries. However, psychological factors, scientific and technological level and political relations are not direct affects the evolution of the food trade networks between China, the United States, Russia, European Union and African countries. They indirectly affects the evolution of the food trade networks between China, the United States, Russia, European Union and African countries through markets, prices, imports, and exports. More importantly, data on psychological factors, scientific and technological level, political relations, and other related indicators of African countries cannot be obtained at present. Therefore, we cannot directly add a discussion on the impact of psychological factors the evolution of the food trade networks between China, the United States, Russia, European Union and African countries. Thanks again for your guidance.
Comments 4/5: The Quadratic Assignment Procedure (QAP) is an extension of the Mantel test within a regression framework, developed in the field of psychometrics.Thus, it is necessary to investigate psychosocial factors as they can aid in the analysis of interrelated items and contribute to social network analysis.
Reponses 4/5: Thanks very much for your advice. You have a very deep understanding of QAP, and we are very inspired by it. We agree with you that it is necessary to conduct an analysis of the impact of psychological factors on the evolution of the food trade network between China, the United States, Europe, the European Union, and African countries. We are preparing to conduct a survey on psychological factors in African countries, and will explore the impact of psychological factors on the evolution of food trade networks in African countries such as China, the United States, Russia, and the European Union in our next paper after data collection is complete. However, we have not yet completed a social survey on psychological factors and are unable to include them in the analysis of driving factors at this time. Thank you very much for your understanding. We truly appreciate the advice you have provided.
Comments 6: The conclusion and discussion should be presented under separate headings.
Reponses 6: Thanks very much for your for your guidance. We have presented separate conclusions and discussions in the revised manuscript.
Comments 7: The discussion should be linked to the trade networks of food in the specified countries.
Reponses 7: Thanks very much for your advice. We strongly agree with your suggestion that the discussion should be linked to the trade networks of food in the specified countries. We have added discussions the specified countries in the discussion section of the food trade network. Please refer to the revised manuscript for details. Thank you once again sincerely.

Reviewer 2 Report
Comments and Suggestions for Authors
The main research objective of this paper is to analyze the evolution of food trade networks from a comparative perspective, specifically focusing on key countries such as China, the United States, Russia, the European Union, and African nations. The study aims to reveal the differences in food trade links and the structural characteristics of these networks, thereby providing insights into the dynamics of food trade and its implications for food security. The objective is clearly formulated. However, it could be improved by explicitly stating the significance of understanding these networks in the context of global food security challenges. The research does address a research gap by adopting a micro perspective that contrasts with previous studies that often focus on macro variables, so no further improvements are required.
The article contains a scientific literature analysis, particularly in the introduction section, where it reviews existing studies on the evolution of food trade networks and the factors influencing these networks at a global level. While the literature analysis provides a foundation for the research, its sufficiency can be questioned. The article could benefit from a more comprehensive review of recent studies that specifically address food trade networks, particularly those that include African countries.
The research methodology employed in the article, including Social Network Analysis (SNA) and Quadratic Assignment Procedure (QAP) analysis, is robust and well-suited for the research objectives. These methodologies provide a strong framework for analyzing the complex relationships within food trade networks. However, I suggest the authors include a longitudinal study that could help understand how food trade networks evolve, particularly in response to global events such as economic crises or climate change. This would further enhance the robustness of the findings.
The article's main conclusions revolve around the evolution of food trade networks among China, the United States, Russia, the European Union, and African countries. These conclusions are based on the findings derived from the regression analysis and social network analysis conducted in the study. The authors provide statistical evidence to support their claims, demonstrating the relationships between various factors and food trade dynamics. However, I recommend the authors to discuss the potential implications of the findings for food security policy, particularly in African countries. This would help to contextualize the research within real-world challenges and potential solutions, making the audience feel the practical relevance of the research. Moreover, including specific recommendations for future research could enhance the conclusions.
To sum up, I recommend that the authors improve the article, considering the recommendations above.
Author Response
Dear reviewer:
Thank you very much for your careful guidance on the paper. Your feedback has greatly helped improve the quality of our paper, and we have greatly benefited from it. According to your comments, we have made a comprehensive revision of the paper. The specific revision parts have been marked in blue in the revised manuscript. We truly appreciate you again.
Comments 1: The main research objective of this paper is to analyze the evolution of food trade networks from a comparative perspective, specifically focusing on key countries such as China, the United States, Russia, the European Union, and African nations. The study aims to reveal the differences in food trade links and the structural characteristics of these networks, thereby providing insights into the dynamics of food trade and its implications for food security. The objective is clearly formulated. However, it could be improved by explicitly stating the significance of understanding these networks in the context of global food security challenges. The research does address a research gap by adopting a micro perspective that contrasts with previous studies that often focus on macro variables, so no further improvements are required.
Reponses 1: Thank you very much for your comments on the article. The revised manuscript has strengthened the background introduction of global food security challenges in the introduction section. Thank you again.
Comments 2: The article contains a scientific literature analysis, particularly in the introduction section, where it reviews existing studies on the evolution of food trade networks and the factors influencing these networks at a global level. While the literature analysis provides a foundation for the research, its sufficiency can be questioned. The article could benefit from a more comprehensive review of recent studies that specifically address food trade networks, particularly those that include African countries.
Reponses 2: Thank you very much for your advice. The article has been revised and improved based on your suggestions, especially by adding recent literature on global food trade networks and food security of African countries. Thank you again.
Comments 3: The research methodology employed in the article, including Social Network Analysis (SNA) and Quadratic Assignment Procedure (QAP) analysis, is robust and well-suited for the research objectives. These methodologies provide a strong framework for analyzing the complex relationships within food trade networks. However, I suggest the authors include a longitudinal study that could help understand how food trade networks evolve, particularly in response to global events such as economic crises or climate change. This would further enhance the robustness of the findings.
Reponses 3: Thank you very much for your advice. Your understanding of QAP is very profound, and we are greatly inspired. This paper collected the data of food trade network and driving factors between China, the United States, Russia, European Union, and African countries in 2001, 2011 and 2021, and carried out QAP analysis of these three cut-off years. However, the data on the food trade network for all years from 2001 to 2021 is incomplete, and there are many missing driving factor data in QAP analysis in some years, making it difficult to conduct QAP analysis on the impact of driving factors on the food trade network for each year. Therefore, it is not possible to conduct longitudinal studies to analyze the impact of economic crises or climate change on food trade networks. In the future, we will conduct a social survey on food security in African countries to overcome the limitation of current driving factor data not covering all years. After completing the social survey, we will further discuss the evolution of the food trade network and the impact of global events such as economic crises or climate change on the food trade network. Thank you again.
Comments 4: The article's main conclusions revolve around the evolution of food trade networks among China, the United States, Russia, the European Union, and African countries. These conclusions are based on the findings derived from the regression analysis and social network analysis conducted in the study. The authors provide statistical evidence to support their claims, demonstrating the relationships between various factors and food trade dynamics. However, I recommend the authors to discuss the potential implications of the findings for food security policy, particularly in African countries. This would help to contextualize the research within real-world challenges and potential solutions, making the audience feel the practical relevance of the research. Moreover, including specific recommendations for future research could enhance the conclusions.
Reponses 4: Thank you very much for your advice. Based on your suggestion, we have added a discussion section to the revised manuscript and analyzed potential implications of the findings for food security policy on African countries in the discussion section. In addition, we added a discussion on specific recommendations for future research at the end of the discussion section. Thank you again.

Reviewer 3 Report
Comments and Suggestions for Authors
In this paper, the Evolution of Grain Trade Network from a Comparative Perspective: An Examination of China, the United States, Russia, European Union and African Countries was investigated. The subject is important in the field and some comments are as following:
1-Please explain in details the QAP (A Quality Assurance Plan (QAP) is a document that outlines the processes and procedures necessary to ensure that a product or service meets the defined standards of quality. It typically includes the establishment of quality objectives, the assignment of responsibilities, the documentation of procedures, and the implementation of a process for continuous improvement. The QAP is designed to prevent defects and ensure quality throughout the production or service delivery process.
2-Please present the Indicators such as weighted degree, weighted out degree, weighted in degree, close ness centrality, and intermediary centrality provide insights into the impact of various countries on the food trade network between China, the United States, Russia, European Union, and African countries from different perspectives in mathematical forms for easy understanding
3-Line 417-420. The authors wrote” These factors include differences in per capita 417 GDP, total population, the proportion of agricultural employment, food production, arable land area for food, renewable freshwater resources per capita, geographical distance between capitals, geographical borders, and institutional distance (Table 1)”. I think the most important factor is political relationships among countries, so can you explain why do you ignoring such factor.
4-Please provide in table some numerical data for the Explanatory variables which appeared in Table 1.
5-The authors wrote” The regression coefficient in 2021 indicates that a 1% increase in the difference in per capita GDP leads to a 0.393% increase in food trade volume. Please give reasons.
6-Line 491: The authors wrote” with similar agricultural technology levels”. Please indicate what are those agricultural technology levels with references.
7-Line 492: what do you mean with Institutional distance.
8-Line 494: renewable freshwater resources per capita. Please indicate what are renewable freshwater resources per capita for each country with references.
9-Sensitivity analysis required.
10- Both figures and English need improvements.
Author Response
Dear reviewer:
Thank you very much for your careful guidance on the paper. Your feedback has greatly helped improve the quality of our paper, and we have greatly benefited from it. According to your comments, we have made a comprehensive revision of the paper. The specific revision parts have been marked in blue in the revised manuscript. We truly appreciate you again.
Comments 1: Please explain in details the QAP (A Quality Assurance Plan (QAP) is a document that outlines the processes and procedures necessary to ensure that a product or service meets the defined standards of quality. It typically includes the establishment of quality objectives, the assignment of responsibilities, the documentation of procedures, and the implementation of a process for continuous improvement. The QAP is designed to prevent defects and ensure quality throughout the production or service delivery process.
Reponses 1: Thank you very much for your question.Your understanding deviates from the QAP model in this article. QAP analysis is not a Quality Assurance Plan, it is Quadratic Assignment Procedure QAP) . QAP is a method used to analyze social network data. It performs regression analysis by comparing the similarity of the grid values of two matrices and provides the correlation coefficient between the two matrices. QAP analysis does not require variables to be independent of each other and does not need to consider the issue of multicollinearity among influencing factors, making the regression analysis of relational data more stable.Therefore, this study selected QAP analysis to analyze the impact of various driving factors on the grain trade network.
Comments 2: Please present the Indicators such as weighted degree, weighted out degree, weighted in degree, closeness centrality, and intermediary centrality provide insights into the impact of various countries on the food trade network between China, the United States, Russia, European Union, and African countries from different perspectives in mathematical forms for easy understanding
Reponses 2: Thanks very much for your advice. We believe that the suggestion is very reasonable, According to your suggestion, we have added an explanation of the indicators of social network analysis in the revised manuscript. Please refer to Table 1 for specific explanations
Comments 3: Line 417-420. The authors wrote” These factors include differences in per capita 417 GDP, total population, the proportion of agricultural employment, food production, arable land area for food, renewable freshwater resources per capita, geographical distance between capitals, geographical borders, and institutional distance (Table 1)”. I think the most important factor is political relationships among countries, so can you explain why do you ignoring such factor.
Reponses 3:Your question is a very good one. Political relationships among countries are indeed factors that affect the evolution of the food trade networks between China, the United States, Russia, European Union and African countries. However, political relations are not direct affects the evolution of the food trade networks between China, the United States, Russia, European Union and African countries. Political relationships indirectly affects the evolution of the food trade networks between China, the United States, Russia, European Union and African countries through markets, prices, imports, and exports. More importantly, data on political relationships indicators between China, the United States, Russia, European Union and African countries cannot be obtained at present. Therefore, We did not include political relationships in the driving factors for the evolution of food trade networks between China, the United States, Russia, European Union, and African countries. We have added an explanation to this in the revised manuscript. Thanks again for your guidance.
Comments 4: Please provide in table some numerical data for the Explanatory variables which appeared in Table 1.
Reponses 4: Thanks very much for your advice. Because the explanatory variable data is too large and too long, we cannot present all the explanatory variable data in Table 1. We have supplied all the explanatory variable datasets in the attachment. Therefore, please refer to the supplementary file for detailed data on explanatory variables.
Comments 5: The authors wrote” The regression coefficient in 2021 indicates that a 1% increase in the difference in per capita GDP leads to a 0.393% increase in food trade volume. Please give reasons.
Reponses 5: Thank you very much for your question. The greater the difference in per capita GDP between China, the United States, Russia, European Union and African countries, the greater the relative difference in food imports and exports between China, the United States, Russia, European Union and African countries, which is conducive to promoting food trade between between China, the United States, Russia, European Union and African countries. According to the OAP user manual (https://www.stata.com/meeting/1nasug/simpson.pdf and https://ideas.repec.org/p/boc/asug01/1.2.html), regression coefficients reflect the degree of influence of influencing factors on the explained variable. The regression coefficient of the impact of the difference in per capita GDP on the the explained variable (food trade volume) is 0.393. Therefore, a 1% increase in the difference in per capita GDP leads to a 0.393% increase in food trade volume.
Comments 6: Line 491: The authors wrote” with similar agricultural technology levels”. Please indicate what are those agricultural technology levels with references.
Reply: Thank you very much for pointing out this issue. The explanation here is unreasonable, and we have made modifications to it. Please refer to the revised manuscript for specific modifications.
Comments 7: Line 492: what do you mean with Institutional distance.
Reponses 7:Thank you for your question. Institutional distance is the difference between countries in rules, norms, regulations, and cognition,which can be obtained by calculating the Worldwide Governance Indicators(Please refer to “The impact of institutional distance on the international diversity–performance relationship”). This article takes institutional distance as a variable that affects the food trade network, which can reveal the impact of institutional distance on the food trade networks between China, the United States, Russia, European Union and African countries. We have added an explanation of institutional distance in the revised manuscript.
Comments 8: Line 494: renewable freshwater resources per capita. Please indicate what are renewable freshwater resources per capita for each country with references.
Reponses 8: Thanks very much for your advice. Renewable internal freshwater resources flows refer to internal renewable resources (internal river flows and groundwater from rainfall) in the country. The data of renewable freshwater resources per capita is sourced from the World Bank database (https://data.worldbank.org.cn/indicator/ER.H2O.INTR.PC?view=chart). Because the data of renewable freshwater resources per capitais is too long, we were unable to directly present the data of renewable freshwater resources per capitais for all countries in the article. We have supplied all the explanatory variable datasets in the attachment. Therefore, please refer to the supplementary file for detailed data on renewable freshwater resources per capita for each country.
Comments 9: Sensitivity analysis required.
Reponses 9: Thanks very much for your advice. Sensitivity analysis is to identify the sensitive factors that have a significant impact on the economic performance indicators of investment projects from multiple uncertain factors one by one. This article conducted correlation analysis and regression testing in QAP analysis, and there is no uncertainty or endogeneity problimem with the driving factors. Therefore, sensitivity analysis is not necessary in this article.
Comments 10: Both figures and English need improvements.
Reponses 10: Thank you very much for your reminder. We have made improvements to the figures and english in the revised manuscript.

Round 2
Reviewer 1 Report
Comments and Suggestions for Authors
Thank you for addressing the comments. This version shows significant improvement. However, a few additional points still need to be addressed to further enhance the quality.
· The introduction section should be divided into two separate sections: 1. Introduction and 2. Literature Review to improve clarity and organization.
· Improve the visibility of Figures 2, 3, and 4 for better clarity.
· Refine the conclusion to provide a concise summary.
· Additionally, create a separate section to address the study's limitations and propose future research directions based on these findings and limitations.
Author Response
Dear reviewer:
Your comments are of great help in improving our paper.Thank you very much for your careful guidance on the paper. According to your comments, We have further improved the manuscript. The specific revision parts have been marked in blue in the revised manuscript. We truly appreciate you again.
Comments 1:The introduction section should be divided into two separate sections: 1. Introduction and 2. Literature Review to improve clarity and organization.
Reponses 1:Thanks very much for your advice.Your suggestion is very good. According to your suggestion, we have divided the introduction section into two separate parts (1. Introduction and 2. Literature Review) in the revised manuscript.
Comments 2:Improve the visibility of Figures 2, 3, and 4 for better clarity.
Reponses 2:Thanks very much for your advice. According to your suggestion, we have improved the visibility of Figures 2, 3, and 4 for better clarity. Please refer to the revised manuscript for details. Thank you very much again.
Comments 3:Refine the conclusion to provide a concise summary.
Reponses 3:Thanks very much for your advice. According to your suggestion, We have condensed the conclusion and abstract of the article. Please refer to the revised manuscript for details. Thank you very much again.
Comments 4:Additionally, create a separate section to address the study's limitations and propose future research directions based on these findings and limitations.
Reponses 4:Thanks very much for your advice. In the last paragraph of the discussion section, we explained the limitations of this study and pointed out future research directions based on these findings and limitations.
Reviewer 3 Report
Comments and Suggestions for Authors
Good morning. The authors replayed on all comments. Although the data on political relationships indicators between China, the United States, Russia, European Union and African countries cannot be obtained at present, but the manuscript is acceptable to be published in Foods
Author Response
Dear reviewer:
Your comments are of great help in improving our paper.Thank you very much for your careful guidance on the paper. According to your comments, We have further improved the manuscript. The specific revision parts have been marked in blue in the revised manuscript. We truly appreciate you again.
Comments 1: Good morning. The authors replayed on all comments. Although the data on political relationships indicators between China, the United States, Russia, European Union and African countries cannot be obtained at present, but the manuscript is acceptable to be published in Foods
Reponses 1: Thank you very much for your guidance on the paper. The lack of analysis on the impact of political relations on the food trade network between China, the United States, Russia, European Union and African countries is a limitation of this study. We have explained this in the discussion section. In the future, we will collect data on political relations indicators between China, the United States, Russia, European Union, and African countries through social surveys, and then analyze the impact of political relationships on the food trade network between China, the United States, Russia, European Union, and African countries. Thank you very much again.